# Dysfunctional TRPM8 signalling in the vascular response to environmental cold in ageing

Dibesh Thapa[1], João de Sousa Valente[1†], Brentton Barrett[1†], Matthew John Smith[1†], Fulye Argunhan[1], Sheng Y Lee[1,2], Sofya Nikitochkina[1], Xenia Kodji[1,3], Susan D Brain[1*]

[1]Section of Vascular Biology and Inflammation, School of Cardiovascular Medicine and Sciences, BHF Centre of Research Excellence, King's College London, London, United Kingdom; [2]Cancer Research UK, Cambridge Institute, University of Cambridge, Cambridge, United Kingdom; [3]Skin Research Institute, Agency of Science, Technology, and Research (A*STAR), Singapore, Singapore

**Abstract** Ageing is associated with increased vulnerability to environmental cold exposure. Previously, we identified the role of the cold-sensitive transient receptor potential (TRP) A1, M8 receptors as vascular cold sensors in mouse skin. We hypothesised that this dynamic cold-sensor system may become dysfunctional in ageing. We show that behavioural and vascular responses to skin local environmental cooling are impaired with even moderate ageing, with reduced TRPM8 gene/protein expression especially. Pharmacological blockade of the residual TRPA1/TRPM8 component substantially diminished the response in aged, compared with young mice. This implies the reliance of the already reduced cold-induced vascular response in ageing mice on remaining TRP receptor activity. Moreover, sympathetic-induced vasoconstriction was reduced with downregulation of the $\alpha_{2c}$ adrenoceptor expression in ageing. The cold-induced vascular response is important for sensing cold and retaining body heat and health. These findings reveal that cold sensors, essential for this neurovascular pathway, decline as ageing onsets.

**\*For correspondence:**
sue.brain@kcl.ac.uk

†These authors contributed equally to this work

**Competing interest:** The authors declare that no competing interests exist.

## Editor's evaluation

The authors have demonstrated that the vascular response to cooling deteriorates as a function of age and that impaired function of cold receptors (viz, TRPM8 and to a lesser extent TRPA1) as well as dysfunctional sympathetic signaling contribute to the attenuated vascular response. These findings importantly add to the understanding of the regulation of vascular tone during physiological aging.

## Introduction

Upon exposure to cold, depending on the type and intensity, several counterbalancing responses are produced, such as shivering thermogenesis involving skeletal muscle, or non-shivering thermogenesis in brown adipose tissue (BAT) and peripheral vasoconstriction in skin (*Señarís et al., 2018*, Morrison, Shaun F., *Morrison and Nakamura, 2011*; *Morrison and Nakamura, 2019*). To produce such responses, thermo-sensors in the form of temperature-sensitive sensory receptors are distributed throughout the skin and are considered to work as a first line of defence against cold, which makes peripheral cutaneous responses a fundamental event in the defence against environmental thermal challenge. The sensory receptors in the skin initiate the vascular cold constrictor response

which acts to protect against body heat loss and prevent hypothermia. This response is followed by the subsequent vasodilation, a restorative response that is essential to protect the affected skin against cold-induced conditions, such as chilblains, trench foot, frostbite, and Raynaud's condition (*Daanen and van der Struijs, 2005*; *Keatinge, 1957*; *Lewis, 1930*). It is a finely tuned well-balanced response that maintains cellular function and physiological homeostasis during cold exposure. Whilst this response is relevant to all ages, physiological changes in ageing leads to dysfunctional signalling which causes a reduced adaptation to cold exposure (*Guergova and Dufour, 2011*). With the lack of physical activity in the elderly population, it exacerbates the fall in core body temperature which can cause fatal cardiovascular and respiratory problems (*Billeter et al., 2014*; *Stares and Kosatsky, 2015*). This is normally the biggest cause behind the National Health Service (NHS) excess winter deaths that we witness every year, where in 2018 it caused approximately 11,000 deaths linked to cold exposure in England (*Office for National Statistics, 2019*).

We have previously delineated the primary roles of transient receptor potential (TRP) channels in producing a distinctive biphasic vascular response to cold in the mouse paw consisting of a TRP ankyrin 1 (TRPA1)/ melastatin 8 (TRPM8)-initiated sympathetic $\alpha_{2c}$ adrenoceptor mediated neuronal vasoconstriction and a distinct TRPA1-CGRP (Calcitonin gene-related peptide) mediated sensory-vasodilator component (*Aubdool et al., 2016*). TRPA1 is a biomolecular sensor for reduced temperatures especially noxious cold ( < 18 °C), mediating aversive behaviour such as avoiding cold-induced pain, whilst also being involved in mediating inflammatory pain (*Kwan et al., 2006*; *Nassini et al., 2014*; *Jain et al., 2011*; *Gouin et al., 2017*). Additionally, it activates C and Aδ sensory nerves to release neuropeptides such as CGRP to mediate neurogenic vasodilation (*Aubdool et al., 2016*; *Story et al., 2003*; *Gentry et al., 2010*). TRPM8 on the other hand is sensitive to low or reduced temperatures such as cool temperatures ( < 28 °C) (*McKemy et al., 2002*; *Peier et al., 2002*). It is involved in deep body cooling and suggested to supersede the role of TRPA1 (*Gavva et al., 2012*). TRPM8 is also suggested to be a vasoactive stimulus (*Bautista et al., 2007*; *Johnson et al., 2005*; *Silva et al., 2019*). The other established receptor that plays a pivotal role in cold signalling is the sympathetic $\alpha_{2c}$ adrenoceptor, which mediates the vasoconstriction of the blood vessels (*Bailey et al., 2004*). Whilst the sympathetic branch that is involved in the vasoconstrictor component of the cold response has been shown to have reduced activity in ageing humans (*Holowatz et al., 2010*; *Degroot and Kenney, 2007*), little is known about the functionality of the cold receptors TRPA1 and TRPM8 in ageing. In the current study, we hypothesize that signalling via the cold receptors TRPA1 and TRPM8 deteriorates with ageing which causes an impaired vascular response to the cold.

The primary objective of this study is to investigate the cutaneous vascular response to cold in ageing, focusing on the activity of the cold TRP receptors; TRPA1 and TRPM8. As sympathetic-sensory neuronal signalling is key for the cutaneous vascular cold response in ageing, we also searched for evidence of dysfunction within these systems. Here using in vivo, ex vivo, genetic, and pharmacological approaches, we show that TRPA1 and TRPM8 signalling declines with ageing which affects the sensing as well as functional pathways involved in cold signalling; all of which contribute to the impaired cold vascular response. Additionally, we provide evidence that the $\alpha_{2c}$ adrenoceptor as well as the TRPM8 receptor both play critical roles to influence this outcome, as the expression of both diminishes significantly in ageing which impacts the vascular response to cold. These important findings establish the dynamic role of cold sensitive TRP receptors and sympathetic receptors in the cutaneous vascular response to the cold as ageing occurs.

## Results

### Cold-induced vascular response is impaired in ageing

We analysed the cold induced vascular response in wild-type (WT) CD1 females (Young: 2–3 months, Aged: 13–15 months) with a full-field laser perfusion imager (FLPI) using the cold water immersion model (*Figure 1a*) developed in our laboratory (*Aubdool et al., 2014*; *Pan et al., 2018*). After the baseline blood flow was measured for 5 min, the ipsilateral hindpaw was immersed in cold water at 4 °C, a temperature that produces a robust vascular response, for 5 min and blood flow was then recorded for another 30 min. The cold treatment produced a typical vascular response of rapid vaso-constriction followed by a prolonged recovery vasodilator response in both young and aged mice (*Figure 1b–c*, *Figure 1—figure supplement 1a*). In young mice, the cold treatment produced a

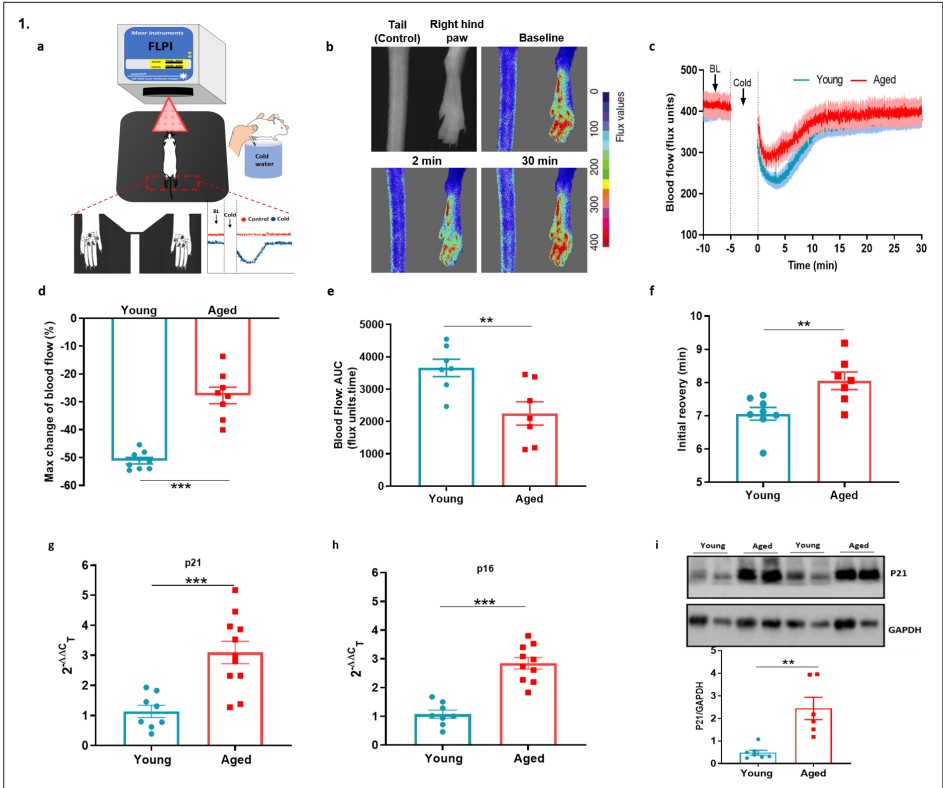

**Figure 1.** Cold-induced vascular response is impaired with ageing. (**a**) Diagram illustrates the experimental setup of cold-induced vascular response protocol; FLPI from top measures the blood flow in the hindpaw of the anaesthetised mouse when on a heating mat in response to cold water immersion. The expanded component (highlighted by dotted red lines) shows the hindpaw region in which the blood flow is recorded, and a graph of typical blood flow response is shown. Recording is paused for cold treatment where one of the hindpaw is immersed in cold water for 5 min. (**b**) Representative FLPI image shows the blood flow in cold-treated hind paw at baseline, 2 min and 30 min after the cold water treatment. (**c**) Graph shows the raw blood flow trace (mean ± s.e.m.) of vascular response with cold (4 °C) water treatment (n = 8). (**d**) % change in hindpaw blood flow from baseline to 0–2 min following cold water treatment (maximum vasoconstriction). (**e**) The AUC to maximum vasoconstriction point assessed by area under the curve (AUC). (**f**) Time of blood flow recovery immediately after maximum vasoconstriction until the start of the plateau period. (**g–h**) RT-PCR CT analysis shows fold change of *P21* and *P16* gene expression normalised to three housekeeping genes in dorsal root ganglia (DRG) of young and aged mice. (**i**) Representative western blot of p21 in hindpaw skin of young and aged mice and densitometric analysis normalised to GAPDH. (BL = baseline). Data is presented as mean and all error bars indicate s.e.m. (n = 6–11) **p < 0.01, ***p < 0.001. (Two-tailed Student's t-test).

The online version of this article includes the following figure supplement(s) for figure 1:

**Figure supplement 1.** Analysis of cold-induced blood flow in the mouse paw.

maximum vasoconstriction of 51.1% ± 1.1%, however, in aged mice this was significantly blunted with maximum vasoconstriction of 27.7% ± 3.0% (*Figure 1d*). These changes were reflected in the area under the response curve (AUC) analysis with a significantly greater response in young than aged mice (*Figure 1e*). The result was extended by measurement of the blood flow recovery after the cold treatment. Although blood flow did not fully recover back to the baseline, the initial rate of recovery immediately after maximum vasoconstriction before it slowly plateaued off was significantly faster in the young mice compared to the aged mice (*Figure 1f*). These results suggest that with ageing the cold induced vascular response starts to diminish, which affects both parts of the vascular response. We were surprised that these changes were observed with moderately aged mice, equivalent to middle aged in human terms (*Dutta and Sengupta, 2016*). However, at this age there is a clear evidence of elevated gene expression in dorsal root ganglion (DRG) and skin of senescence markers associated

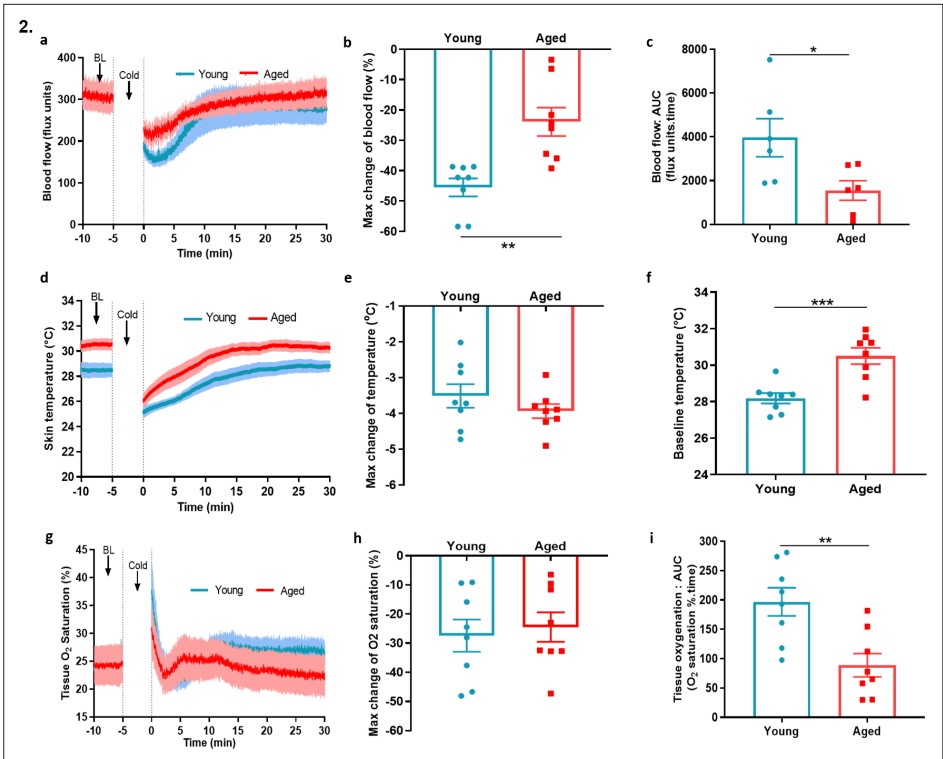

**Figure 2.** Blood flow, skin temperature and tissue oxygen saturation with cold treatment in ageing. (**a**) The mean ± s.e.m. blood flow trace of the vascular response with cold (4 °C) water treatment (n = 8). (**b**) % change in hindpaw blood flow from baseline to 0–2 min following cold treatment (maximum vasoconstriction). (**c**) The vasoconstriction response caused by cold water treatment represented by area under curve (AUC). (**d**) The mean ± s.e.m. recordings of hindpaw skin temperature with cold water treatment. (**e**) Maximum reduction in skin temperature following 5 min cold treatment. (**f**) The baseline skin temperature. (**g**) % mean ± s.e.m. tissue oxygen saturation during cold water treatment. (**h**) % maximum change in tissue oxygen saturation from baseline following cold water treatment (**i**) % tissue oxygen saturation recovery after cold water treatment assessed by area under the curve. (BL = baseline). Data is presented as mean and all error bars indicate s.e.m. (n = 8) *p < 0.05, **p < 0.01, ***p < 0.001. (Two-tailed Student's t-test).

The online version of this article includes the following figure supplement(s) for figure 2:

**Figure supplement 1.** Oxidative stress with ageing.

with ageing, *P16; INK4A, cyclin-dependent kinase inhibitor 2* A (*P16*) and *P21; Waf1, cyclin-dependent kinase inhibitor 1* (*P21*), (*Figure 1g–h*) also confirmed by western blotting (*Figure 1i*).

To extend our mechanistic understanding, we also used a laser Doppler imager (VMS-LDF), in addition to FLPI, which simultaneously measures the blood flow, skin temperature and tissue oxygen saturation level at a single point, to investigate the vascular response to cold. Similar to the results obtained using the FLPI, the environmental cold water treatment produced an impaired vascular response in the paws of aged mice compared to the young mice (*Figure 2a*). In young mice, the cold treatment produced a maximum vasoconstriction of 45.5% ± 3.0%; however, in aged mice, this was significantly lower with a maximum vasoconstriction of 23.4% ± 4.7%, a result which was reflected in the AUC analysis (*Figure 2b–c*). There was a trend of greater reduction in skin temperature of aged mice after the cold-water treatment; however, the aged mice had a significantly higher skin temperature at baseline. This suggests that they were losing more body heat and is consistent with the fact that the ability to maintain core body temperature declines with ageing (*Figure 2d–f*). The tissue oxygen saturation level underwent a similar reduction in both young and aged mice after the cold exposure (*Figure 2g–h*) but recovered more robustly in the young mice compared to the aged mice as shown by AUC analysis (*Figure 2i*). We also found evidence of increased cellular stress as protein expression of 3-nitrotyrosine (3-NT), a biomarker of oxidative stress produced via reactive nitrogen species was elevated in aged hindpaw skin (*Figure 2—figure supplement 1*), in keeping

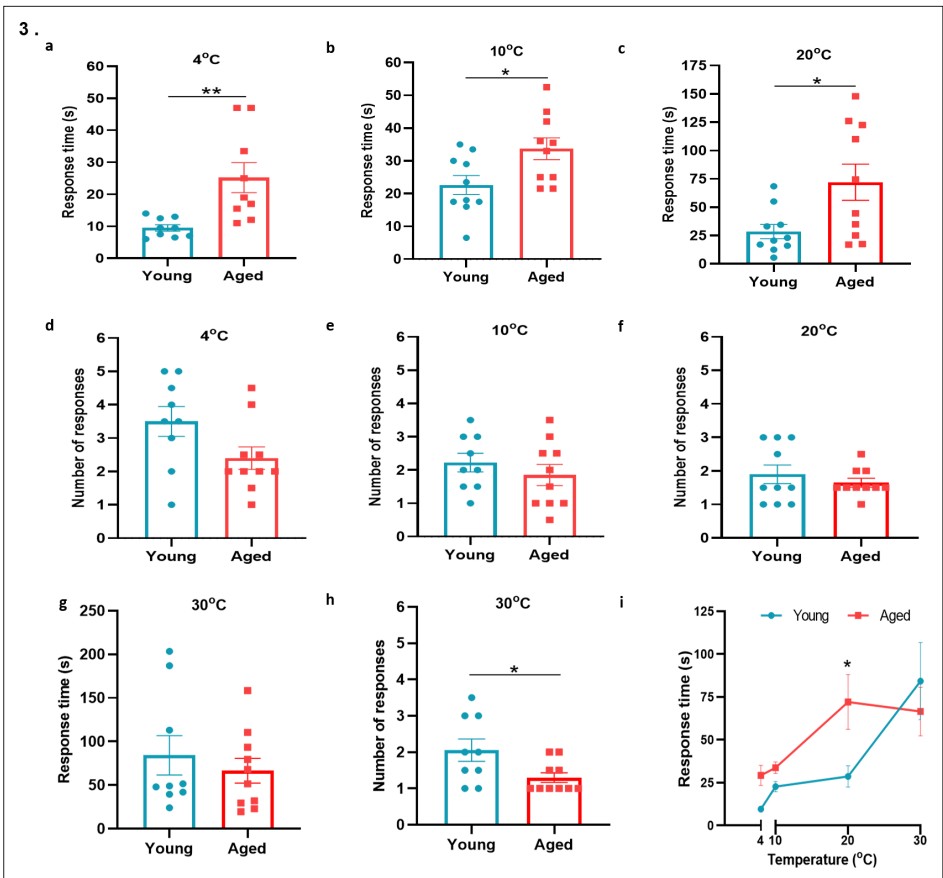

**Figure 3.** Behavioural analysis with cold plate in young and aged mice. (**a–c**) Time of first response of mice to cold plate set at 4 °C, 10 °C, and 20 °C. (**d-f**) The total number of responses from mice during the cold plate experiment at 4 °C, 10 °C, and 20 °C. (**g–h**) Time of first response of mice to cold plate set at 30 °C and total number of responses. (**i**) Line graph illustrates the difference in mean response time at the four different temperatures the cold plate assay was performed, between young and aged mice. All results are shown as mean ± s.e.m. (n = 9–10) *p < 0.05, **p < 0.01. (Two-tailed Student's t-test or [Two-way ANOVA with Tukey's post hoc test]).

with physiological ageing. These results from two distinct techniques confirm our finding that the cold induced vascular response starts to diminish with ageing.

## Cold sensitivity is impaired in ageing

To learn if cold sensitivity had altered with ageing, we examined the functionality of TRPA1 and TRPM8 channels in behavioural studies using a cold plate set at 4 °C, 10 °C, and 20 °C, within the activation range of TRPA1 and TRPM8 receptors (*Dhaka et al., 2007*; *Kwan et al., 2006*). At all three low/reduced temperatures, the aged mice showed a significant delayed latency for paw licking/paw withdrawal/jumping compared to the young mice, suggesting impaired cold sensing in aged mice (*Figure 3a–c*), but with little difference in the total number of responses observed among groups (*Figure 3d-f*). When the test was performed at 30 °C, a temperature outside the activation range of TRPA1 and TRPM8, we observed no delayed latency in response time, although the total number of responses was significantly lower in the aged mice (*Figure 3g–h*). These results indicate that there is a reduction in sensitivity to cold with ageing at temperatures at which the cold sensors TRPA1 and TRPM8 are active; thus leading us to hypothesise that at least one cold-sensitive TRP pathways deteriorates with ageing. Of note, a secondary analysis of this data, where this data is placed in a summary figure (*Figure 3i*) is carried out. Here, the response time was compared with the overall temperatures tested. The largest difference in response time compared with temperature between young and aged mice was observed at 20 °C, in keeping with the TRPM8 activation range (*Figure 3i*).

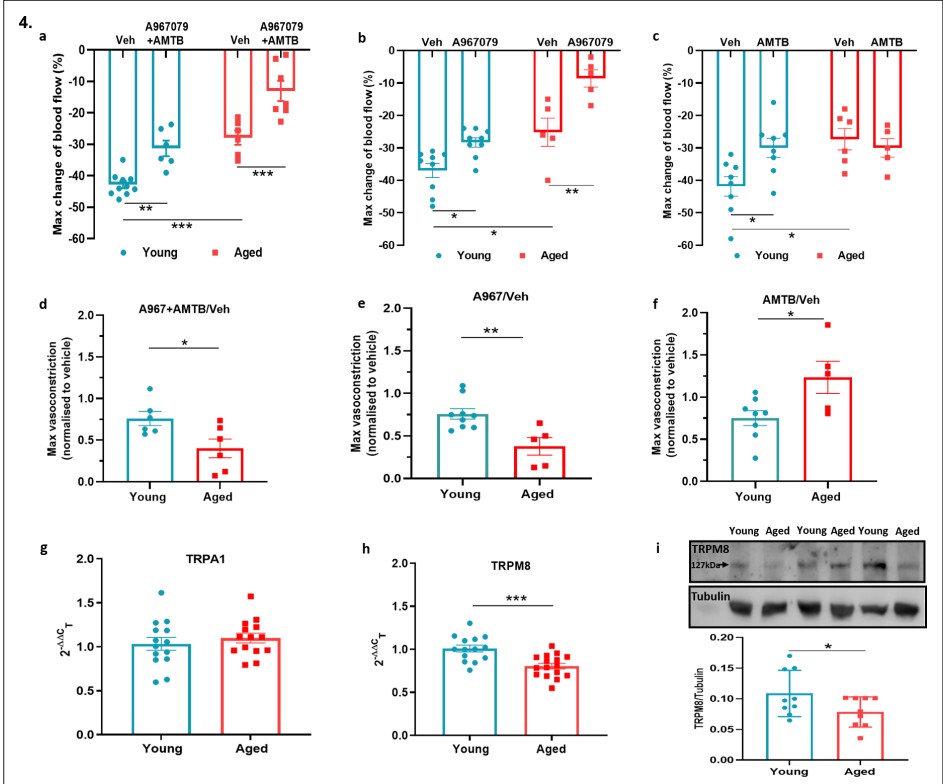

**Figure 4.** TRPA1 and TRPM8 are involved in cold-induced vascular response. Vascular responses with cold (4 °C) water treatment in mice pre-treated with combined TRPA1 antagonist A967079 (100 mg kg$^{-1}$) and TRPM8 antagonist AMTB (10 mg kg$^{-1}$), or vehicle control (Veh - 10 % DMSO, 10 % Tween in saline) i.p. 30 min before cold treatment. (**a–c**) % change in hindpaw blood flow from baseline to 0–2 min following cold treatment (maximum vasoconstriction) in mice treated with combined antagonist (**a**) A967079+ AMTB, (**b**) A967079, and (**c**) AMTB. (**d-f**) Maximum vasoconstriction caused by cold water treatment in mice treated with combined antagonist (**d**) A967079+ AMTB, (**e**) A967079, and (**f**) AMTB normalised against vehicle treated mice. (n = 5–10) (**g–h**) RT-PCR CT analysis shows fold change of (**g**) *Trpa1* and (**h**) *Trpm8* normalised to three housekeeping genes in DRG. (n = 14–15) (**i**) Representative western blot of TRPM8 in DRG of young and aged mice and densitometric analysis normalised to Tubulin (Y = young, A = aged) (n = 9). All results are shown as mean ± s.e.m. *p < 0.05, **p < 0.01, ***p < 0.001. (Two-way ANOVA with Tukey's post hoc test or Student's t-test).

## The cold-induced vascular response remains dependent on TRPA1 but not TRPM8 in ageing

To investigate the role of TRPA1 and TRPM8 in the local cold water immersion test; we measured the cold-induced vascular response in the presence of the TRPA1 antagonist A967079 (100 mg kg$^{-1}$ intraperitoneal [i.p.]) and TRPM8 antagonist AMTB (10 mg kg$^{-1}$ i.p.), a combination previously shown by us to inhibit the cold induced vascular response (*Pan et al., 2018*). The combined pre-treatment of A967079 and AMTB partially but significantly inhibited the vasoconstriction in young mice. By comparison, this treatment regime produced a more substantial inhibition of vascular responses induced by cold in the aged mice (*Figure 4a and d*). This result reveals that the role of TRP receptors in the cold-induced vascular response remains and suggests as ageing occurs the TRP-mediated signalling may become more important. Next, we performed the cold water immersion test in the presence of either A967079 or AMTB. The A967079 treatment produced a similar effect to the combined antagonist treatment of A967079+ AMTB, where the antagonist was more effective in aged than in the young mice (*Figure 4b and e*). By comparison, the AMTB treatment inhibited the response in young mice, but had no significant effect in the vascular response to cold (*Figure 4c and f*) in aged mice. This provides further evidence that as ageing occurs, TRPM8 loses its ability to respond to local cold treatment. Next, we examined the expression of TRPA1 and TRPM8 in DRGs of young and aged mice. RT-PCR analysis of DRG showed similar level of *Trpa1* mRNA in both young and aged mice

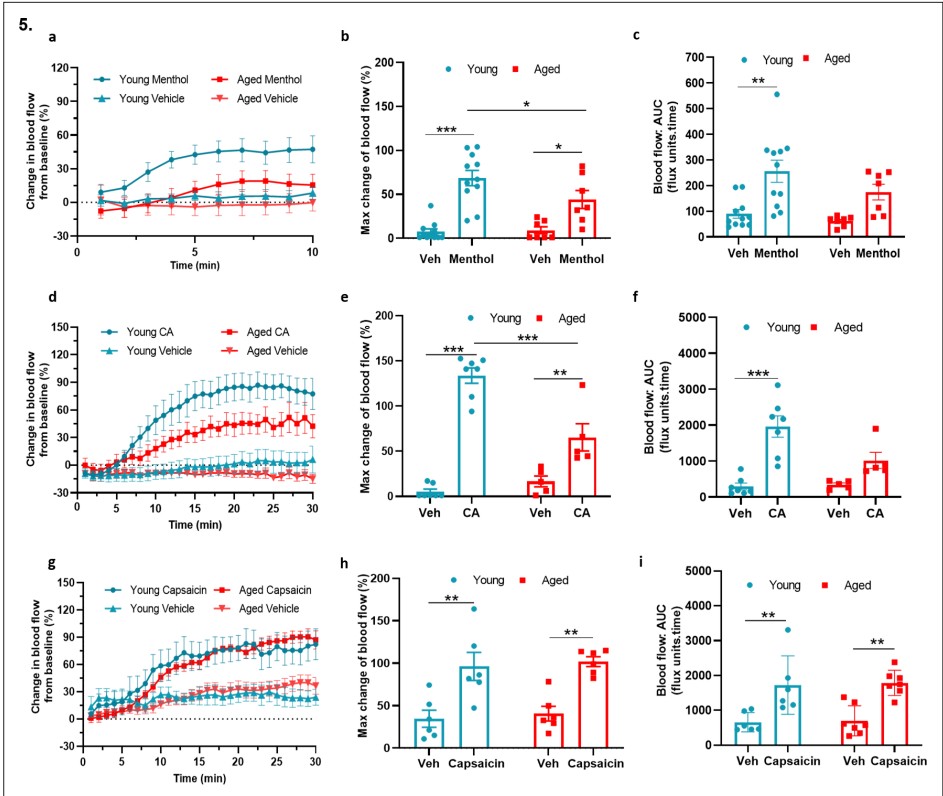

**Figure 5.** TRPA1 and TRPM8 activity deteriorates with ageing (**a**) Graph shows the % mean ± s.e.m. of blood flow change from baseline in response to topical application of menthol (10%) and vehicle (Veh - 10 % DMSO in ethanol) in ear of young and aged mice. (**b**) % maximum change in ear blood flow induced by menthol application in young and aged mice. (**c**) AUC analysis of % blood flow increase from baseline after menthol application compared to vehicle. (**d**) Graph shows the % mean ± s.e.m. of blood flow change from baseline in response to topical application of cinnamaldehyde (10 % CA) and vehicle (10 % DMSO in ethanol) in ear of young and aged mice. (**e**) % maximum change in ear blood flow induced by CA application in young and aged mice. (**f**) AUC analysis of % blood flow increase from baseline after CA application compared to vehicle. (**g**) Graph shows the % mean ± s.e.m. of blood flow change from baseline in response to topical application of capsaicin (10%) and vehicle (10 % DMSO in ethanol) in ear of young and aged mice. (**h**) % maximum change in ear blood flow induced by capsaicin application in young and aged mice. (**i**) AUC analysis of % blood flow increase from baseline after capsaicin application compared to vehicle. All results are shown as mean ± s.e.m. (n = 5–11) *p < 0.05, **p < 0.01, ***p < 0.001. (Two-way ANOVA with Tukey's post hoc test).

The online version of this article includes the following figure supplement(s) for figure 5:

**Figure supplement 1.** Agonist induced blood flow response in mouse ear.

---

(**Figure 4g**). We could not further investigate TRPA1 protein expression, due to lack of availability of a TRPA1 antibody with sufficient selectivity. However, the level of *Trpm8* mRNA was significantly reduced in the aged compared to the young mice (**Figure 4h**), as was its protein expression when analysed by western blot (**Figure 4i**).

## TRPA1 and TRPM8 vasodilator signalling is impaired in ageing

Thus far, we had gained multiple evidence that TRPM8 activity is impaired in vascular signalling in ageing, with some evidence for a reduction in TRPA1 activity. To build on these findings, we examined the vasoactive effect of TRPA1 and TRPM8 agonists that are commonly associated with sensory nerves. The topical application of cinnamaldehyde (CA) and menthol on mouse skin have previously been shown to mediate vasodilation via TRPA1 and TRPM8 channels, respectively (**Craighead et al., 2017**; **Aubdool et al., 2016**). The topical application of menthol (10%) to the ear caused increased blood flow in young mice, which was significantly lower in the aged mice (**Figure 5a**), as shown by the maximum increase in blood flow (**Figure 5b**). The AUC analysis of blood flow showed a significant

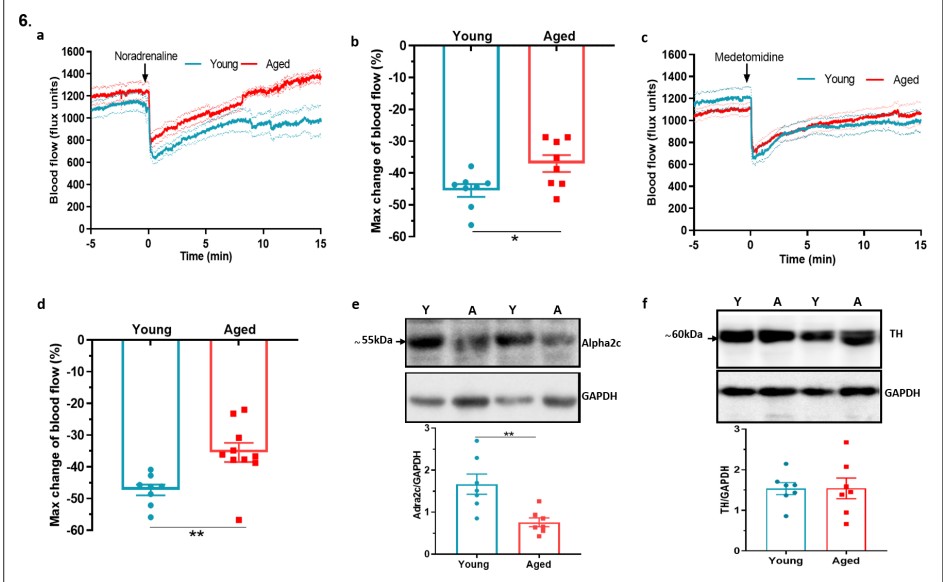

**Figure 6.** Dysfunctional sympathetic signalling in ageing (**a**) Graph shows the mean ± s.e.m. blood flow in hindpaw with intraplantar injection of noradrenaline (1.25 ng/µl in saline in 20 µl) in young and aged mice (n = 8). (**b**) % maximum change in blood flow from baseline induced by noradrenaline (maximum vasoconstriction). (**c**) Graph shows the mean ± s.e.m. blood flow in hindpaw with intraplantar injection of medetomidine (1.25 ng/µl in saline in 20 µl) in young and aged mice (n = 8–10) (**d**) % maximum change in blood flow from baseline induced by medetomidine (maximum vasoconstriction). (**e**) Representative western blot of alpha2C ($\alpha_{2c}$) adrenoceptor in mice hindpaw skin with densitometric analysis normalised to GAPDH. (**f**) Representative western blot of tyrosine hydroxylase (TH) in mice hindpaw skin with densitometric analysis normalised to GAPDH (Y = young, A = aged). (n = 7) *p < 0.05, **p < 0.01. (Two-tailed Student's t-test).

The online version of this article includes the following figure supplement(s) for figure 6:

**Figure supplement 1.** Phospho-tyrosine hydroxylase with ageing.

increase with menthol treatment compared to vehicle in young mice, but not in aged mice (*Figure 5c*). Similarly, CA (10%) application also increased blood flow in young mice, however, this increase was significantly lower in aged mice (*Figure 5d–e*, *Figure 5—figure supplement 1*). The AUC analysis showed a significant increase in blood flow with CA treatment compared to vehicle in young mice but not in aged mice (*Figure 5f*). These findings suggest that the TRPA1 and TRPM8-mediated vasoactive activity starts to deteriorate in moderate ageing, and it is not exclusive to cold signalling. To build on this concept, we examined whether the activity of another prominent TRP receptor, TRP vanilloid 1 (TPRV1), is also impaired with ageing. To probe this, we studied capsaicin-induced increase in ear blood flow (*Grant et al., 2005*). The topical application of 10 % capsaicin produced a similar increase in ear blood flow in both young and aged mice (*Figure 5g–i*) indicating, unlike TRPA1 and TRPM8 signalling, TRPV1 signalling does not deteriorate with ageing.

## Dysfunction in sympathetic signalling contributes to impaired cold response in ageing

In comparison to the sensory system, the importance of sympathetic nerves in mediating the vascular smooth muscle constriction in the cold response is well established (*Bailey et al., 2004*; *Smith et al., 2004*). To understand whether there is modulation of this pathway as ageing progresses, we examined the sympathetic-mediated vasoconstriction. The response to the intraplantar injection of the non-selective and endogenous sympathetic neurotransmitter noradrenaline (NA) revealed a significantly greater reduction of blood flow in young mice compared to aged mice (*Figure 6a–b*). Knowing that the $\alpha_{2c}$ adrenoceptor is essential for cold induced vasoconstriction (*Aubdool et al., 2014*; *Bailey et al., 2004*; *Honda et al., 2007*), we then proceeded to investigate the effect of the selective $\alpha_2$ adrenoceptor agonist medetomidine, in hindpaw blood flow. Medetomidine caused immediate vasoconstriction as expected, but the response was blunted in aged mice compared to young mice

(*Figure 6c–d*). These results recapitulate previous findings that suggest a defect also in sympathetic signalling in aged mice involving the $\alpha_{2c}$ adrenoceptor, in addition to the cold TRP receptors. The western blotting analysis of the hind paw skin showed a significant reduction in the expression of $\alpha_{2c}$ adrenoceptor in aged mice (*Figure 6e*). To elucidate further potential defects in the sympathetic pathway with ageing, we investigated the biosynthesis pathway of NA, the major signalling molecule of sympathetic system. Tyrosine hydroxylase (TH), an enzyme that catalyses the rate limiting stage of noradrenaline synthesis, showed a similar level of expression (*Figure 6f*) including of its active form, phosphorylated TH in both young and aged mice (*Figure 6—figure supplement 1*), suggesting the production of noradrenaline remained unaltered with ageing. This indicates that in ageing the expression and function of the $\alpha_{2c}$ adrenoceptor diminishes and that contributes to the impaired constrictor response against cold.

## Sympathetic-sensory signalling and influence of ageing

We extended our investigation of sympathetic system in vascular cold response by exploring potential crosstalk between sympathetic and sensory signalling in ageing. To elucidate this, we first investigated DRG and found that $\alpha_{2a}$ and $\alpha_{2cc}$ *adrenoceptor* gene expression was reduced in ageing (*Figure 7—figure supplement 1a-b*) similar to that shown for the *Trpm8* gene, (*Figure 4h*) whilst no significant difference was found for *Trpv1* receptors (*Figure 7—figure supplement 1c*) in keeping with results for *Trpa1* (*Figure 4g*). By comparison, whilst the TRP receptors are well known to be expressed in sensory neurons there is evidence for a broader localisation (*Hirai et al., 2018*; *Jain et al., 2011*; *Smith et al., 2004*; *Yang et al., 2006*). We investigated the possible expression of these receptors on sympathetic nerves by collecting the sympathetic ganglia (SG) from the cervical and thoracic paravertebral regions where they could directly influence the NA transmission that mediates the vasoconstrictor component of the vascular cold response. To confirm the phenotype of sympathetic neurons, we used positive markers such as tyrosine hydroxylase (TH) and dopamine β-hydroxylase (Dbh) (*Figure 7—figure supplement 2a-b*) both of which exhibited high expression compared to sensory neuron of DRGs and kidney which were used as negative controls. The RT-PCR data on SG showed the gene expression of both *Trpa1* and *Trpm8* in young and aged mice. Interestingly, the expression of both receptors were significantly downregulated in aged mice (*Figure 7a–b*). Whilst there is no feasible selective TRPA1 antibody available, western blot analysis of TRPM8 on SG recapitulated the qPCR finding of diminished expression in aged mice compared to young mice (*Figure 7c*). These findings reveal expression of cold TRP receptors in sympathetic neurons which are diminished in ageing.

To confirm whether the expression of TRPA1 and TRPM8 receptors were functionally active, we investigated the ability of TRPM8 and TRPA1 agonists to activate SG cultured cells using calcium imaging. The TRPM8 agonist menthol (300 µM) caused an increase in calcium as measured by the change in fura-2 fluorescence ratio compared to vehicle (*Figure 7d*). This response was significantly reduced in the presence of the TRPM8 antagonist AMTB (10 µM) (*Figure 7d–e*). Similarly, the TRPA1 agonist Allyl isothiocyanate (AITC) (200 µM) also caused an increase in calcium in these cultured cells compared to vehicle (*Figure 7f*), which was significantly inhibited by the TRPA1 antagonist A967079 (10 µM) (*Figure 7f–g*). These results show that TRPM8 and TRPA1 receptors are expressed in sympathetic neurons and activate sympathetic nerves.

$\alpha_{2c}$ – alpha2c adrenoceptor, $\alpha_{2a}$ – alpha2a adrenoceptor, VOCC- voltage operated calcium channel, NA – noradrenaline, $Ca^{2+}$ - calcium, SMC- smooth muscle cell, EC – endothelial cell.

## Discussion

The role of TRPA1 and TRPM8 as cold-sensitive thermoreceptors is established (*Story et al., 2003*; *Bautista et al., 2007*; *Karashima et al., 2009*; *Peier et al., 2002*; *McKemy et al., 2002*) and our research has demonstrated the essential role they play as vascular cold sensors (*Aubdool et al., 2014*; *Pan et al., 2018*). Much less was known about their activity in ageing, until this study. We provide a new insight into the changing roles of TRPA1 and TRPM8 in the vascular response to cold in ageing; the expression and activity of TRPM8 is significantly diminished, and to a lesser extent TRPA1-mediated signalling too.

The vascular response to the cold is a primary physiological response, which we have previously teased out the key mechanisms for, consisting of TRPA1/M8 initiated $\alpha_{2c}$-mediated sympathetic

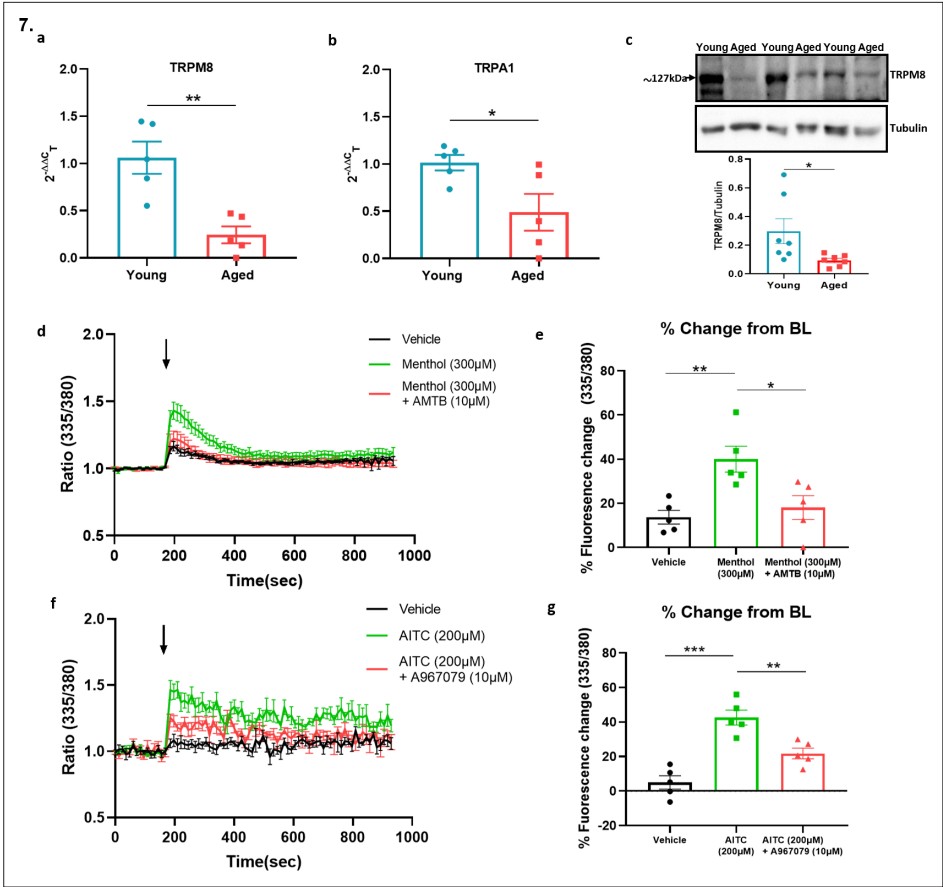

**Figure 7.** Sympathetic-sensory signalling and influence of ageing. (**a–b**) RT-PCR CT analysis shows the expression and fold change of *Trpa1* and *Trpm8* in young and aged SG normalised to three housekeeping genes collected from the cervical and thoracic paravertebral region. (**c**) The western blot analysis of TRPM8 in SG of young and aged mice. All results are shown as mean ± s.e.m. (n = 5–7) *p < 0.05, **p < 0.01. (Two-tailed Student's t-test). (**d**) Shows the raw trace of intracellular calcium change (mean ± s.e.m.) induced by menthol (300 μM) shown by the ratiometric fura-2 fluorescence change over time. (**e**) Maximum % change in fluorescence change from baseline caused by menthol in the presence and absence of AMTB (10 μM). (**f**) Shows the raw trace of intracellular calcium change (mean ± s.e.m.) induced by AITC (200 μM) shown by the ratiometric fura2 fluorescence change over time. (**g**) Maximum % change in fluorescence change from baseline caused by AITC in the presence and absence of A967079 (10 μM). Arrow represents the time-point of agonist injection. n = 5 independent experiments, where each treatment was a mean of replicate values. All results are shown as mean ± s.e.m. *p < 0.05, **p < 0.01, ***p < 0.01. (One-way ANOVA with Tukey's *post hoc* test or Two-tailed Student's t-test).

The online version of this article includes the following figure supplement(s) for figure 7:

**Figure supplement 1.** Sympathetic-sensory signalling in young and aged mice.

**Figure supplement 2.** Characterisation of sympathetic markers.

vasoconstriction followed by TRPA1/M8-mediated sensory vascular relaxation after localised cold exposure in the mouse paw. Here, we show that the response is functionally deficient in ageing as measured by two different laser blood flow measurement techniques. Both components of the cold-induced vascular response were impaired in ageing with blunted vasoconstriction that will lead to increased heat loss, and a slower rate of recovery that may lead to cold-induced injuries (*Keatinge, 1957*; *Roustit et al., 2011*; *Herrick, 2005*). We were surprised that the diminished response was observed with even moderate ageing (13–15 months old mice; equivalent to middle age in humans). However, the ageing nature of the mice was confirmed by the observation of increased expression of the established ageing markers p16 and p21 (*Baker et al., 2016*; *Sharpless and Sherr, 2015*; *Hudgins et al., 2018*). This finding is in keeping with the concept that although ageing-induced pathological conditions and frailty appear at a later age, the underlying physiological changes that manifest those

conditions begin at a middle age of around 40 years old in humans. Indeed, this is when the brain volume and weight start to reduce *Peters, 2006*; cardiovascular functions begin to decline and elite athletes start to lose stamina (*Pal et al., 2014*; *Mühlberg and Platt, 1999*). We found that the baseline skin temperature was significantly higher in the aged mice than young mice, potentially due to greater heat loss from aged mice in keeping with the notion that it is harder to maintain core body temperature as ageing occurs. The tissue oxygen saturation was reduced during the vascular cold response but recovered substantially in the young compared to the aged mice. Overall, the findings show that the cold-induced cutaneous vascular response is significantly diminished in ageing.

It is known that sensory modalities decline with ageing, but usually studies involve frail 24 -month-old mice. Indeed, in one of the only studies of TRP thermo-receptors in ageing, to our knowledge, authors investigated the changes in TRPM8 expressing neurons in cornea with ageing and its relevance to dry eye disease (*Alcalde et al., 2018*). Here in this study, we delineate the activity and expression of one of the prominent sensory cold channels (TRPM8) diminishes with moderate ageing; relevant to the impaired vascular response to cold that we have observed. We designed experiments to evaluate the ability of the mouse to sense cold using a cold plate behavioural assay at innocuous cool (TRPM8 range) and noxious cold (TRPA1 range) temperatures. We observed a delayed latency to the cold response in moderately aged mice compared to young mice at three different cold temperatures, 4 °C, 10 °C, and 20 °C, implying impaired sensory signalling of TRPM8 and TRPA1 receptors. Importantly, the longest delayed latency was observed at 20 °C, which falls under the TRPM8 activation range implying that with ageing TRPM8 signalling deteriorates more, relative to that of TRPA1.

Using combined selective antagonists of TRPA1 and TRPM8, we show that blocking both receptors inhibits the cold-induced cutaneous vascular response in young mice, as expected. Intriguingly, the same treatment produced a relatively stronger inhibitory response in the aged mice. This implies that with ageing the cold signalling relies profoundly more on cold TRP channels. These results may also indicate that at younger ages other protein/s besides TRPA1 and TRPM8 play a crucial part in the cold signalling, but these activities begin to decline with ageing. This includes the $\alpha_{2c}$ receptor as discussed below, although a range of other candidates have also been proposed (*Zimmermann et al., 2011*; *Noël et al., 2009*; *Luiz et al., 2019*; *Gong et al., 2019*). Importantly, when we investigated the effect of the TRPA1 antagonist alone, we observed a very similar inhibitory profile to that of the combination of TRPA1 and TRPM8 antagonists. However, the TRPM8 antagonist treatment alone, was effective in the young mice, but not in the aged mice. This provided key evidence that the activity of TRPM8 especially, is diminished in ageing. TRPM8 was discovered as a sensory receptor expressed in DRG and trigeminal ganglia (TG) that is activated by cool temperatures ( < 28 °C) *McKemy et al., 2002*; although the link with TRPA1 containing CGRP fibres is less well defined (*Hondoh et al., 2010*; *Kobayashi et al., 2005*). Since then, various reports have suggested that TRPM8 is expressed in a wider range of tissues and is involved in multiple physiological functions including thermoregulation (*Moraes et al., 2017*; *Hirai et al., 2018*; *Yang et al., 2006*). Indeed, it is established that the deletion/ antagonism of TRPM8 increases heat loss and reduces core body temperature (*Almeida et al., 2012*; *Reimúndez et al., 2018*).

By this stage, we had revealed a reduced expression and activity of TRPM8 in the vascular cold response and cold sensing. However, we have previously defined TRPA1 as an essential vascular sensor to cold, playing a major role in the cold induced vascular response alongside TRPM8, which we considered to be involved, but not in such an essential manner (*Aubdool et al., 2014*). Therefore, it was surprising to observe that expression of TRPA1, unlike TRPM8, did not diminish in the DRG with ageing, especially as the cold-sensing data from the cold plate at noxious cold temperatures revealed that the response is also impaired at noxious temperatures in ageing. To learn more, we studied the ability of the TRPA1 agonist cinnamaldehyde (CA) to increase cutaneous blood flow; as TRPA1 is localised to CGRP[+] sensory nerves (*Aubdool et al., 2016*). CA-induced vasodilation was significantly impaired in the aged mice compared to young, supporting the concept of impaired functional TRPA1 vascular responses in ageing. We observed a similar significantly blunted response with the TRPM8 agonist menthol in aged mice, although TRPM8 localisation to sensory nerves is more limited than that of TRPA1 (*Kobayashi et al., 2005*). This led us to question whether activity of all TRP receptors deteriorates with ageing, through investigating the non-cold sensing TRPV1 agonist capsaicin which activates predominantly CGRP[+] C-fibres (*Story et al., 2003*; *Sharrad et al., 2015*). In contrast to menthol and CA, capsaicin caused a similar level of increased blood flow in all mice regardless of

age indicating TRPV1 activity does not deteriorate with ageing, and supporting our behaviour data at 30 °C, which falls outside TRPA1 and TRPM8 activation ranges. These results suggest that only the signalling of cold TRP receptors; TRPA1 and TRPM8 is impaired with ageing.

The cold-induced vasoconstriction phase is mediated by sympathetic drive comprising of noradrenergic nerves and this signalling has been shown to decline with ageing (*Degroot and Kenney, 2007*; *Frank et al., 2000*; *Greaney et al., 2015*). Thus, we aimed to elucidate the sympathetic signalling in young and aged mice, which we began by investigating the effect of exogenous agonist NA. NA administered locally to the paw evoked cutaneous vasoconstriction in young mice that was significantly blunted in the aged mice, suggesting that NA-mediated response diminishes in aged mice. Nonetheless, NA is a non-selective agonist for all adrenoceptors, but peripheral cutaneous vasoconstriction is mediated via α adrenergic receptors (*Drew and Whiting, 1979*), with cold-specific vasoconstriction primarily mediated via $\alpha_{2c}$ adrenoceptors subtype (*Bailey et al., 2004*; *Honda et al., 2007*). Therefore, we used the selective $\alpha_2$ agonist medetomidine which induced vasoconstriction that was also significantly blunted in the aged mice. The result suggests that either $\alpha_{2c}$ receptor sensitivity declines with ageing (*Thompson et al., 2005*) or $\alpha_{2c}$ receptor number reduces with ageing which has been suggested to occur in ageing human saphenous vein (*Hyland and Docherty, 1985*). In our study, we found a significant reduction in the expression of $\alpha_{2c}$ adrenoceptors. We also investigated whether the level of NA or its synthesis was impaired in ageing and observed no difference in the level of tyrosine hydroxylase (including the active form of phosphorylated tyrosine hydroxylase), the enzyme involved in the rate limiting synthesis of NA production. This indicates that NA synthesis is not affected in ageing.

The cold-induced vascular response is perceived as a reflex where peripheral sensory nerves sense the cold stimulus and send information to the central nervous system (CNS). In turn, the CNS processes the information and produces an appropriate response via activation of sympathetic nerves to cause vasoconstriction in skin (*Chotani et al., 2000*). Classically, it is established that sensory receptors TRPA1 and TRPM8 that sense cold reside in sensory nerves and alpha-adrenergic receptors

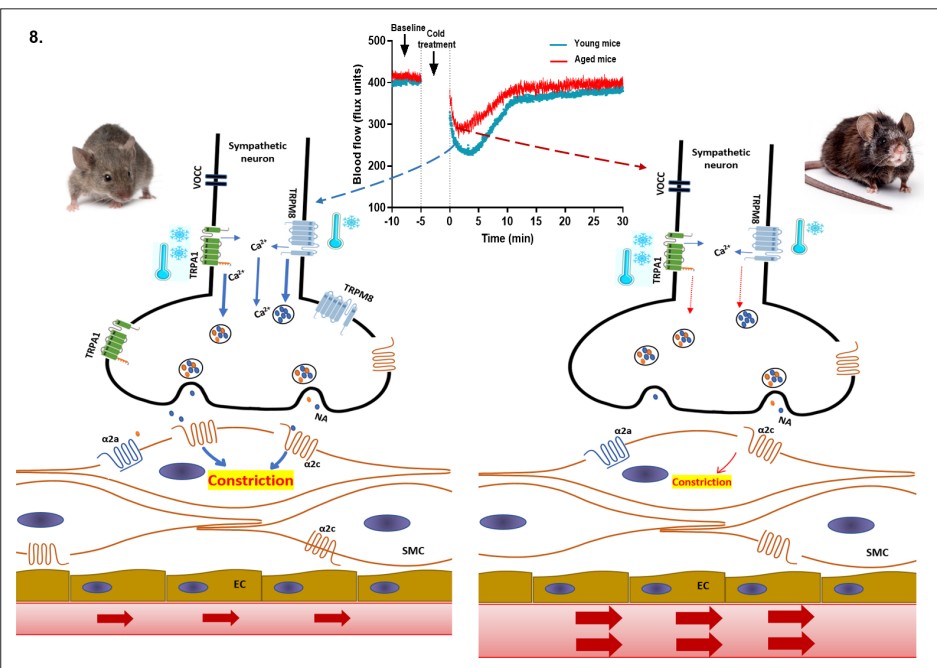

**Figure 8.** Proposed cold-induced vasoconstriction signalling pathway in young and aged mice. The local cold exposure produces rapid vasoconstriction which is significantly blunted in the aged mice (see blood flow graph at top centre). Cold water (4 °C) exposure to hindpaw activates the cold receptors TRPA1 and TRPM8 in sympathetic nerves, which leads to increased intracellular calcium and release of NA. This signalling, however, is significantly downregulated in aged mice due to diminished expression of TRPA1/TRPM8 in sympathetic nerves. NA acts on the post-synaptic $\alpha_{2c}$ adrenergic receptors on smooth muscle cells to mediate vasoconstriction. However, $\alpha_{2c}$ adrenergic receptor are also significantly diminished in aged mice, which leads to reduced signalling. All these factors contribute to an attenuated vascular cold response in aged mice compared to young mice.

reside in sympathetic nerves and smooth muscle cells to modulate vasoconstriction. However, we have previously shown that the cold-induced vasoconstrictor response occurs as normal, when the sensory C-fibre component is removed with resiniferatoxin treatment (*Aubdool et al., 2014*). We interpret this as the vasodilator component not affecting the constrictor component in young mice at least, as the vasoconstrictor component occurs before the relaxant component is activated. This clear result raises the possibility that the cold-sensitive proteins TRPA1 and TRPM8 may be expressed in other tissues besides sensory nerves and modulate the vascular tone, as suggested to be the case in some organs (*Earley, 2012*; *Johnson et al., 2005*; *Yang et al., 2006*). The sensory nerves and sympathetic nerves are known to have close proximation around blood vessels and have a reciprocal trophic influence (*Terenghi et al., 1986*). Thus, we questioned whether cold TRP channels were expressed on sympathetic nerves to directly modulate the vascular tone. We found that both TRPA1 and TRPM8 are expressed in the SG collected from the cervical and thoracic paravertebral regions, in keeping with previous studies (*Smith et al., 2004*), but debated. Furthermore, the expression of both receptors were significantly diminished in the SG collected from the aged mice compared to young mice. To determine if the TRPA1 and TRPM8 receptors can activate mouse sympathetic neurons, we used a calcium imaging assay. Cultured SG were collected from naïve young WT mice. We show that the TRPA1 agonist AITC and the TRPM8 agonist menthol both elicited calcium influx into the cultured cells. Importantly a 10 min pre-treatment of these cells with the TRPA1 antagonist A967079 and the TRPM8 antagonist AMTB significantly inhibited the calcium response induced by AITC and menthol, respectively.

Collectively, these findings suggest that cold stimuli activate TRPA1 and TRPM8 channels on sympathetic nerves, which induces calcium-dependent release of vesicles containing NA into the synaptic cleft where they activate the $\alpha_{2c}$ adrenoceptor on smooth muscle cells to mediate vasoconstriction (*Figure 8*). This signalling cascade has been shown in PC12 cells, an in vitro model for sympathetic neurons (*Smith et al., 2004*; *Peixoto-Neves et al., 2018*; *Yoshimura et al., 2016*). It indicates a potential for sympathetic-sensory interactive signalling in skin, which weakens as ageing progresses, in turn affecting the sensitivity of the vascular response to cold.

To conclude, we have revealed that the cold induced defensive responses decline with ageing. There is an impairment in the sympathetic vasoconstrictor pathway concomitant with a functional deterioration and molecular loss of TRPM8 and TRPA1 signalling as well as diminished $\alpha_{2c}$ adrenergic receptor expression and activity. We consider that the finding of diminished TRPM8 expression with ageing is indicative of a major influence of this channel that would lead to the impaired response to cold observed in ageing, such as in the cold induced vascular response.

## Materials and methods

**Key resources table**

| Reagent type (species) or resource | Designation | Source or reference | Identifiers | Additional information |
|---|---|---|---|---|
| Other | CD1 | Charles River | 022; RRID:IMSR_CRL:22 | Mouse (*Mus musculus*) |
| Antibody | Anti-TRPM8 (Rabbit polyclonal) | Alomone Labs | Cat# ACC-049, RRID:AB_2040254 | "(1:500)" |
| Antibody | Anti-alpha2c adrenergic receptor (Rabbit polyclonal) | Novus Biologicals | Cat# NB100-93554, RRID:AB_1236641 | "(1:1000)" |
| Antibody | Anti-tyrosine hydroxylase (phospho ser41) (Rabbit polyclonal) | Novus Biologicals | Cat# NB300-173, RRID:AB_2201648 | "(1:1000)" |
| Antibody | Anti-tyrosine hydroxylase (Rabbit polyclonal) | Novus Biologicals | Cat# NB300-109, RRID:AB_10077691 | "(1:1000)" |
| Antibody | Anti-p21 (Mouse monoclonal) | Santa Cruz Biotechnology | Cat# sc-6246, RRID:AB_628073 | "(1:1000)" |
| Antibody | Anti-GAPDH (Rabbit polyclonal) | Thermo Fisher Scientific | Cat# PA1-987, RRID:AB_2107311 | "(1:2000)" |

*Continued on next page*

*Continued*

| Reagent type (species) or resource | Designation | Source or reference | Identifiers | Additional information |
|---|---|---|---|---|
| Antibody | Anti- β- actin (Mouse monoclonal) | Sigma-Aldrich | Cat# A5441, RRID:AB_476744 | "(1:2000)" |
| Antibody | Anti- α - tubulin (Rat monoclonal) | Millipore | Cat# MAB1864, RRID:AB_2210391 | "(1:2000)" |

## Animals

Female CD1 mice used in this study were either bred in the Biological Services Unit, King's College London or purchased from Charles River (Kent, UK). The animals were housed in a climatically controlled environment with an ambient temperature of 22 °C, including a 12 hr light/dark (7am-7pm) cycle with free access to drinking water and standard chow ad libitum. Young mice were 2–3 months old and aged mice were 13–16 months old. All experiments were performed according to the Animal Care and Ethics committee at King's College London, in addition to the regulations set by the UK home office Animals (Scientific Procedures) act 1986. Experiments using animals were designed and reported in line with the ARRIVE guidelines, which form the NC3Rs initiative. Animals were randomly assigned to different groups and the investigator was blinded to drug treatments and where possible to the age of the animals.

## Cutaneous blood flow measurement by full-field laser perfusion imager

Mice were terminally anaesthetised with i.p. injection of ketamine (75 mg kg⁻¹) and medetomidine (1 mg kg⁻¹). Wherever possible to comply with the NC3Rs reduction guidelines, experiments were designed using recovery anaesthesia, which was either sub cutaneous (s.c.) administration of 150 mg kg⁻¹ ketamine and 4.25 mg kg⁻¹ xylazine, or isoflurane gas inhalation. The choice of anaesthesia was also chosen to fit with the experimental plan, (e.g. study of the ear or the paw was sometimes impossible with isoflurane, due to the fitting of the mask). 5 % isoflurane (in oxygen) was used to induce anaesthesia, which was followed by 2 % for maintenance during the experimental procedure. Full-field Laser Perfusion Imager (FLPI, Moor Instruments, UK) was used to measure blood flow in the hind paw or the ear of the mice. The mice were placed in a ventral position on a heating mat to maintain core body temperature at 37 °C during blood flow measurement. For the cold-induced blood flow measurement in hindpaw, after anaesthesia, the blood flow was measured on the plantar surface of hindpaw for 5 min as baseline measurement. Then, the ipsilateral hindpaw at the level between tibia and calcaneus was immersed in cold water (4 °C for 5 min) for cold exposure. After the cold treatment, mice were placed back on the heating mat (37 °C) to measure blood flow recovery for 30 min. The FLPI uses laser light to produce speckle pattern that is altered by blood flow which is measured as arbitrary flux units (x10³ flux units). For the agonist-induced blood flow measurement in ear, after anaesthesia, the blood flow was measured for 5 min as a baseline recording. 10 µl of either cinnamaldehyde (10 % CA, 10 % DMSO in ethanol), menthol (10 % menthol, 10 % DMSO in ethanol), or capsaicin (10 % capsaicin, 10 % DMSO in ethanol) was topically applied to both sides of the ipsilateral ear and 10 µl vehicle solution (10 % DMSO in ethanol) was applied to the contralateral ear. Then blood flow was measured for 30 min after cinnamaldehyde and capsaicin treatment and for 10 min after menthol treatment. All treatments produced a gradual increase in blood flow. For NA/medetomidine-induced blood flow measurement in hindpaw, after anaesthesia, blood flow was measured on the plantar surface for 5 min as baseline recording. We chose to use isoflurane for this experiment to ensure that medetomidine as part of our anaesthesia did not influence our results. Intraplantar injection of NA/medetomidine (1.25 ng/µl) was performed and blood flow was measured for 15 min.

## Cutaneous blood flow, temperature, and oxygen saturation measurement by laser doppler techniques

A probe connected to the moorVMS-LDF (Laser Doppler Perfusion and Temperature Monitor) and moorVMS-OXY (Tissue Oxygen and Temperature Monitor) all from (Moor Instruments, UK) was used to simultaneously measure blood flow, temperature and tissue oxygen saturation in a small, localised

area (~5 mm diameter) on the plantar surface of the ipsilateral hind paw (central region immediately adjacent to the digits). The probe was held on a retort stand clamp 1 mm above the skin surface. After inducing anaesthesia with i.p. injection of ketamine (75 mg kg$^{-1}$) and medetomidine (1 mg kg$^{-1}$), the blood flow was measured (baseline recording) on the plantar surface central area immediately adjacent to the digits for 5 min. The ipsilateral hindpaw was immersed in cold water (4 °C for 5 min) at the level between tibia and calcaneus. After the cold treatment, mice were placed back on the heating mat (37 °C) to record all measurements during the recovery period for 30 min. The blood flow was measured using doppler technique and expressed in arbitrary flux units, and tissue oxygen saturation was measured using white light spectroscopy method.

## Drugs and reagents

The TRPA1 antagonist A967079 ((1E,3E)–1-(4-Fluorophenyl)–2-methyl-1-pentene-3-one oxime); (Alomone Labs, # A-225) was dissolved in 10 % DMSO (Dimethyl sulfoxide) with 10 % Tween-80 in saline. The TRPM8 antagonist AMTB (N-(3-aminopropyl)–2-[(3-methylphenyl) methyl] oxy-N-(2-thienylmethyl) benzamide hydrochloride salt; Alomone Labs, #A-305) was dissolved in 10 % DMSO in saline. Both antagonists were administered i.p. 30 min before the cold treatment. Cinnamaldehyde (Sigma-Aldrich, #W228613, > 95% purity), menthol (Alfa Aesar, #A18098, 98 % purity), capsaicin (Sigma-Aldrich, #M2028, >95% purity) were prepared with 10 % DMSO in ethanol solution. 1.25 ng/μl NA (Sigma-Aldrich) and 1.25 ng/μl medetomidine (Orion Pharma) were administered via intraplantar injection in 20 μl saline.

## Behavioural testing using the cold plate

The nociceptive cold sensitivity response of mice was tested using a hot/cold thermal plate (Ugo Basile 35100). A quick temperature non-contact infrared thermometer (Linear labs) was used to confirm the set temperature of the plate before each experiment. Prior to the experiments, mice were acclimatised to the room for 30 min for 3 days, and the thermal plate by individually placing them on the plate at room temperature for 2 min on each of the 3 days. At the start of the experiment, the plate was set to the chosen temperature (4 °C, 10 °C, 20°C, and 30°C) and each mouse was placed individually onto the plate in turn. The cold response was detected as either paw licking or paw withdrawal/ jumping and the total number of responses observed within 1 min (for 4°C and 10°C) and 5 min (20°C and 30°C) were tallied. Each temperature was repeated twice on different days to obtain an average which was used to plot the final graph.

## Quantitative polymerase chain reaction

Real time PCR (RT-qPCR) was used to quantify changes in mRNA collected from pooled DRG, and SG collected from thoracic paravertebral region. The total RNA was isolated and purified according to manufacturer's instructions using the RNeasy Micro Kit (Qiagen, #74004). The RNA concentration and absorbance ratio (A260/280 and A260/230) were measured using Nanodrop 2000 spectrophotometer. 500 ng of purified RNA was reverse transcribed using SuperScript ViLO cDNA synthesis kit (Thermo Fisher Scientific, #11754050). qPCR was performed with 10 ng of cDNA using PowerUp SYBR Green master mix kit (Thermo Fisher Scientific, #A25780) in 7900HT Real-Time PCR machine (Applied Biosystems, USA). All primers (*Supplementary file 1*) were designed using Primer-BLAST software (NCBI) according to MIQE guidelines and checked on the primer stat website. (http://www.bioin-formatics.org/sms2/pcr_primer_stats.html). The melting curve analysis was performed after reactions to confirm specificity of the primers. The analysis was performed using delta delta CT method and expressed as fold change normalised to the average of three housekeeping genes.

## Western blotting

The western blotting analysis was performed as previously described (*Aubdool et al., 2014*). The tissue was lysed with sodium dodecyl sulphate (SDS) lysis buffer which was made up with inhibitors of both phosphatases and proteases (1 tablet per 10 ml, #4693159001 + #4906845001, Sigma-Aldrich). The tissue was then homogenised using a tissue lyser (Qiagen, #85300). The protein concentration was determined using the Bradford protein dye binding assay (#5000113 + #5000114, Bio-Rad). Fifty μg of protein was separated by electrophoresis in an SDS-polyacrylamide gel which was then transferred using the semi dry method, onto PVDF membranes. The membrane was incubated in a blocking buffer

made up of 5 % BSA in Phosphate-buffered saline- tween (PBS-T) (0.1 % Tween). The membrane was blocked for 1 hr at room temperature (RT) except for TRPM8 which was blocked for 2.5 hr as per manufacturer's instruction. The membranes with primary antibodies were incubated overnight at 4 °C. Following the washing step with PBS-T, the membranes were probed with secondary antibody (Horse-radish peroxidase conjugated) (1:2000 dilution, #AP132P Sigma-Aldrich) for 1 hr at RT. The enhanced chemiluminescence (ECL, Piercenet) was used for visual development of the membranes inside a gel doc system. Bands were normalised to housekeeping genes *α-tubulin* (1:2000, #MAB1864, Merck Millipore, RRID:AB_2210391), *Gapdh* (1:2000, #PA1987, Thermo Fisher Scientific, RRID:AB_2107311) and *β-actin* (1:2000, #A5441, Sigma Aldrich, RRID:AB_476744). Quantitative western blot analysis was performed using Image J (NIH, USA) software. The primary antibodies were made in 3 % PBST solution at 1:500 dilution for TRPM8 (Alomone Labs #ACC049, RRID:AB_2040254), 1:1,000 dilution for $\alpha_{2c}$ adrenergic receptor (Bio-Techne #NB100-93554, RRID:AB_1236641), phospho TH (Bio-Techne #NB300-173, RRID:AB_2201648), total TH (Bio-Techne #NB300-109, RRID:AB_10077691), p21 (Santa Cruz #sc-6246, RRID:AB_628073).

## Intracellular calcium measurement

SG were isolated from the cervical and thoracic paravertebral region of sympathetic chain from young male mice (8–12 weeks). The ganglia were digested with 0.125 % collagenase (Sigma-Aldrich, #C1889) and separated cells plated onto poly-L-ornithine (Sigma-Aldrich, #P8638) coated 96-well black-walled, clear-bottomed plate (Greiner Bio-One) with Dulbecco's Modified Eagle's Medium/Nutrient Mixture F-12 Ham (Sigma-Aldrich, #D6421), penicillin/streptomycin, 10 % fetal calf serum, and 50 ng/ml NGF (Promega # G5141) for 24 hours. Cells were incubated with 4 µM Fura-2 AM (Invitrogen #ab120873) and 500 µM probenecid (Sigma-Aldrich, #P8761) in serum-free DMEM/F-12 Ham media for 1 hr at 37 °C, 5 % $CO_2$. Cells were then washed twice with PBS before addition of Hank's balanced salt solution (HBSS) (Thermo Fisher Scientific, #14025050). Changes in intracellular calcium levels were then monitored in the entire well using a fluorescence plate reader (CLARIOstar, BMG Labtech) at excitation: 335/380 nm, emission: 510 nm. The agonists AITC (200 µM), menthol (300 µM) or vehicle (DMSO) were added using the plate readers integrated reagent injector system at the stated time-points. The antagonists A967079 (10 µM) and AMTB (10 µM) were pre-incubated for 10 min in the cells before addition of their respective agonist.

## Experimental design and data analysis

The majority of the experiments conducted in this study consisted of two groups (young/aged) or four groups with drug treatments (young/aged and vehicle/drug), therefore the power analysis from our lab (*Aubdool et al., 2014*) with a power of 80 % (0.8) for a confidence of 5 % (0.05) recommended n = 8, which was adhered to where possible. The order of the mice (young or aged) and treatments (vehicle/drug) received were randomised during experimental protocols. Data was analysed using either two-tailed Student's t-tests or two-way ANOVA followed by Tukey's post hoc test. All column data are plotted as dot plots to show variability and n numbers for each data set. All data are expressed as mean ± SEM. $p < 0.05$ was considered to represent a significant difference. GraphPad Prism (version 8) was used as statistics software for analysis.

# Acknowledgements

This work was primarily funded by BBSRC (BB/P005616/1). It was also supported in part by Versus Arthritis (ARUK21524) and British Heart Foundation (BHF- FS/19/42/34527 and PG/12/34/29557). We thank Professor Giovanni Mann for academic input concerning the calcium imaging experiments.

## Additional information

### Funding

| Funder | Grant reference number | Author |
|---|---|---|
| Biotechnology and Biological Sciences Research Council | BB/P005616/1 | Dibesh Thapa |
| Versus Arthritis | ARUK21524 | João de Sousa Valente |
| British Heart Foundation | FS/19/42/34527 | Brentton Barrett |
| British Heart Foundation | PG/12/34/29557 | Fulye Argunhan |

The funders had no role in study design, data collection and interpretation, or the decision to submit the work for publication.

### Author contributions

Dibesh Thapa, Conceptualization, Data curation, Formal analysis, Investigation, Methodology, Supervision, Validation, Writing – original draft, Writing – review and editing; João de Sousa Valente, Data curation, Investigation, Methodology, Validation, Writing – review and editing; Brentton Barrett, Data curation, Formal analysis, Investigation, Methodology, Writing – original draft, Writing – review and editing; Matthew John Smith, Data curation, Formal analysis, Investigation, Methodology, Writing – review and editing; Fulye Argunhan, Data curation, Investigation, Methodology, Writing – review and editing; Sheng Y Lee, Data curation, Formal analysis, Investigation, Writing – review and editing; Sofya Nikitochkina, Data curation, Investigation, Writing – review and editing; Xenia Kodji, Formal analysis, Methodology, Writing – review and editing; Susan D Brain, Conceptualization, Funding acquisition, Investigation, Methodology, Project administration, Resources, Supervision, Validation, Visualization, Writing – original draft, Writing – review and editing

### Author ORCIDs

Dibesh Thapa ⬡ http://orcid.org/0000-0002-7435-5483
Susan D Brain ⬡ http://orcid.org/0000-0002-9684-8342

### Ethics

All experiments were performed according to the Animal Care and Ethics committee at King's College London, in addition to the regulations set by the UK home office Animals (Scientific Procedures) act 1986. The protocol was approved by UK home office under license number P2C5FC8CF. Experiments using animals were designed and reported in line with the ARRIVE guidelines, which form the NC3Rs initiative.

### Decision letter and Author response

Decision letter https://doi.org/10.7554/eLife.70153.sa1
Author response https://doi.org/10.7554/eLife.70153.sa2

## Additional files

### Supplementary files

- Supplementary file 1. List of primer sequences.

- Transparent reporting form

- Source data 1. Orginal blots.

- Source data 2. Raw source files.
 *Figure 1c*: FLPI blood flow data; *Figure 2a*: VMS Blood flow data; *Figure 2d*: Skin Temperature; *Figure 2g*: Oxygen Saturation; *Figure 4a*: BF data with antagonist; *Figure 4b*: BF data A967; *Figure 4c*: BF data AMTB; *Figure 5a*: BF - Menthol; *Figure 5d*: BF - CA; *Figure 5g*: BF – Capsaicin; *Figure 6a*: BF – NA; *Figure 6c*: BF - Medetomidine; *Figure 7d*: Calcium imaging data; *Figure 7f*: Calcium imaging data.

## Data availability

All data generated or analysed during this study are included in the manuscript and supporting files. The source data file with original uncropped western blot images have been uploaded (The blots are labelled as they are in the manuscript). Source data excel file containing raw data for blood flow graphs has also been uploaded which was used for graphical analysis in the main manuscript.

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
