## [Editor Report]

The authors have demonstrated that the vascular response to cooling deteriorates as a function of age and that impaired function of cold receptors (viz, TRPM8 and to a lesser extent TRPA1) as well as dysfunctional sympathetic signaling contribute to the attenuated vascular response. These findings importantly add to the understanding of the regulation of vascular tone during physiological aging.

---

## [Decision Letter]

**Decision letter after peer review:**

Thank you for submitting your article "Dysfunctional TRPM8 signalling in the vascular response to environmental cold in ageing" for consideration by *eLife*. Your article has been reviewed by 2 peer reviewers, and the evaluation has been overseen by a Reviewing Editor and a Senior Editor. The following individual involved in review of your submission has agreed to reveal their identity: Andras Garami (Reviewer #2).

The reviewers have discussed their reviews with one another and this letter summarizes how to prepare your revised submission.

Essential revisions:

1) The authors should provide evidence that the expression of TRPM8 and TRPA1 in sympathetic neurons is associated with clear functional responses. This could be obtained by the assay of NE in skin homogenates after selective TRPM8 or TRPA1 stimulation.

2) Furthermore, the ability of selective agonists to activate (and selective antagonists to inhibit) a calcium-response in cultured sympathetic neurons should be investigated.

3) An additional proof of the exclusive role of TRPM8 in the reduced response to cold could derive by using TRPM8 deleted mice. In these mice aging should not affect the vasoconstrictor response to cold.

4) The proposed hypothesis that the lack of TRPM8 contribution in aged mice is somehow contradicted by the observation that the vasodilating response to menthol was reduced but not absent in aged mice. Given that the vasodilatation elicited by CA is presumably mediated by CGRP, which is the mediator of TRPM8-evoled vasodilatation? Could the reduced vasodilatation that follows TRPM8/TRPA1 activation somehow affect the vasoconstriction? In this line, as TRPA1 on sensory nerve terminal releases CGRP, it would be of interest to exclude that CGRP inhibition does not affect the vasoconstrictor response to cold.

5) It is explained that the cold-induced vasomotor response is present even after the chemical ablation of sensory fibers (ln. 651-653), therefore the age-dependent reduction in expression of TRPM8 and TRPA1 channels develops probably in sympathetic nerves. However, since expression of the channels in sensory nerves was not studied in the aging mice, it cannot be firmly excluded that an age-associated impairment in their cold-detecting function on afferent fibers also contributes to the attenuation of the vascular response. The authors should consider to discuss this possibility.

6) In cases when the changes in a parameter are presented over time (Figures 1c, 2a,d,g, 3i, 5a,d,g, and 6a,c), the authors should consider to run 2-way ANOVA with a post hoc test. Interestingly, the use of ANOVA is described in the text (ln. 791), but the results are not reported. In several of the aforementioned figure panels, the standard errors are missing from the curves, which should be corrected.

7) The authors describe three different methods of anesthesia (ln. 691-696). It should be clarified which method was used in which experiments and, if necessary, explain whether different methods of the anesthesia could lead to differences in the results. A related question requiring explanation is that if medetomidine was used as part of the anesthesia (ln. 692), could it interfere with the effects of intraplantar medetomidine (or noradrenaline) used during the experiments (ln. 714)?

8) Shivering thermogenesis belongs to the autonomic thermoeffectors instead of behavioral responses. The sentence in the Introduction should be corrected accordingly (ln. 54-58).

9) Instead of the expressions "cool/cold temperatures", the authors should consider to use, for example, low/decreased/reduced temperature.

10) The legend of Figure 2d (ln. 345-346) should be revised: "mean blood flow recordings of hindpaw skin temperature" does not seem to make sense.

11) The authors state that brown adipose tissue was collected from the mice for PCR (ln. 750-751), but data are not reported. Please clarify.

12) The use of abbreviations should be revised throughout the manuscript. Some abbreviations are not defined at first mention (e.g., NHS, WT, RT), while others are spelled out multiple times (e.g., CA). The reference to Σ-Aldrich should be unified, because, for example, in the Western blot section three different versions are used.*Reviewer #1 (Recommendations for the authors):*

The authors provide evidence that aged mice show a reduced skin vasoconstrictive response to cold. While in young mice the response is mediated by both TRPM8 and TRPA1, in aged mice TRPA1 was found the exclusive mediator, as TRPM8 component appears to be lost. The authors propose that TRPM8 and TRPA1 that regulate the response to cold are located on terminals of sympathetic nerve fibers where they promote the release of norepinephrine (NE) that by acting on alpha2c adrenoceptors on vessel smooth muscle cells mediate vasoconstriction. Thus, the reduced/absent contribution of TRPM8 attenuates the overall response. Reduction in TRPM8 is sympathetic neurons was shown by western blot, an approach that did not show any change in TRPA1 expression. No change in tyrosine β-hydroxylase suggested that no variation on NE release could be implicated. The sound pharmacological approach provides robust support to the proposed hypothesis. However, additional evidence is required to prove that the sympathetic nerve terminal is the sole anatomical structure that, by a reduced TRPM8 expression/function, determines the attenuated vasoconstrictor response to cold.

*Reviewer #2 (Recommendations for the authors):*

The study by Thapa et al., takes the baton from a series of interesting papers from the same laboratory on the role of TRP channels in vascular responses to cold. The experiments are clear and thoughtful and the presented data support the conclusions. The authors show that the vascular response to cooling deteriorates as a function of age and that impaired function of cold receptors (viz., TRPM8 and to a lesser extent TRPA1) as well as dysfunctional sympathetic signaling contribute to the attenuated vascular response. I have only a few suggestions.

1) It is explained that the cold-induced vasomotor response is present even after the chemical ablation of sensory fibers (ln. 651-653), therefore the age-dependent reduction in expression of TRPM8 and TRPA1 channels develops probably in sympathetic nerves. However, since expression of the channels in sensory nerves was not studied in the aging mice, it cannot be firmly excluded that an age-associated impairment in their cold-detecting function on afferent fibers also contributes to the attenuation of the vascular response. The authors should consider to discuss this possibility.

2) In cases when the changes in a parameter are presented over time (Figures 1c, 2a,d,g, 3i, 5a,d,g, and 6a,c), the authors should consider to run 2-way ANOVA with a post hoc test. Interestingly, the use of ANOVA is described in the text (ln. 791), but the results are not reported. In several of the aforementioned figure panels, the standard errors are missing from the curves, which should be corrected.

3) The authors describe three different methods of anesthesia (ln. 691-696). It should be clarified which method was used in which experiments and, if necessary, explain whether different methods of the anesthesia could lead to differences in the results. A related question requiring explanation is that if medetomidine was used as part of the anesthesia (ln. 692), could it interfere with the effects of intraplantar medetomidine (or noradrenaline) used during the experiments (ln. 714)?

---

## [Author Response]

Essential revisions:1) The authors should provide evidence that the expression of TRPM8 and TRPA1 in sympathetic neurons is associated with clear functional responses. This could be obtained by the assay of NE in skin homogenates after selective TRPM8 or TRPA1 stimulation.

We now provide evidence of the functional expression of TRPM8 and TRPA1 in sympathetic neurons with calcium imaging. We used cultured cells isolated from the sympathetic ganglia (SG) of young WT mice. The cells were stimulated with TRPA1 and TRPM8 agonists (please see answer for question 2 below for further details). We consider this technique (calcium imaging) allows demonstration of the ability of TRPA1 and TRPM8 agonists to directly activate sympathetic neurons to confirm the functional response of these receptors. See revised figure 7 (7d, 7e, 7f, 7g). Dr Matt Smith played an essential role in helping us with these experiments and we add him as an author.

It is noted that we have already investigated noradrenaline levels in a previous study in vivo (Aubdool et al., 2014). We measured noradrenaline concentrations of TRPA1 WT and KO hindpaw tissue collected at the peak (0-2 min) vasoconstriction phase following local cold (10°C for 5 min) water immersion, where cold altered noradrenaline levels in skin from WT but not TRPA1KO mice (Aubdool et al., 2014, Supplementary Figure 5). Contrary to what we predicted, the NA levels were reduced, which we considered due to the intense vasoconstriction and short half-life of noradrenaline (Aubdool et al., 2014). For this reason, we preferred the calcium imaging approach, involving cultured neuronal cells isolated directly from mouse sympathetic ganglia. We were able to directly stimulate these cells with TRPA1 and TRPM8 agonists to confirm their functional activity.

2) Furthermore, the ability of selective agonists to activate (and selective antagonists to inhibit) a calcium-response in cultured sympathetic neurons should be investigated.

We now show this by using the TRPM8 agonist menthol and the TRPA1 agonist allyl isothiocynate (AITC) in the revised Figure 7 (7d-g). Using freshly isolated SG from young CD1 mice, we cultured the cells and performed calcium imaging. We show that TRPA1 and TRPM8 channels are activated on the cultured sympathetic neuronal cells by their respective agonist AITC and menthol, which shows that these receptors are functionally active in sympathetic neurons. Additionally, we have used selective antagonists (A967079 for TRPA1 and AMTB for TRPM8) and found that the respective responses are significantly reduced thus confirming selectivity (Figure 7d-g). Please see new figure 7d-g, the Results section (line 265-274) and Discussion section (lines 721-726) where we have now presented these new findings. The methods are found in lines 860-875.

3) An additional proof of the exclusive role of TRPM8 in the reduced response to cold could derive by using TRPM8 deleted mice. In these mice aging should not affect the vasoconstrictor response to cold.

We agree, this would be a good way forward. However, we are unable to use TRPM8 KO aged mice, as they take much more time to grow than we have available, especially as this colony was frozen down when our animal house was closed at the start of the Covid-19 epidemic and are not up and running yet. On the other hand, we have previously utilised young TRPM8 knockout mice in our Aubdool et al., 2014 manuscript, where they did indeed have a reduced cold-induced vascular cold response. Here the cold-induced vasoconstriction was partially but significantly reduced in TRPM8 KO mice as well as in WT mice pre-treated with the TRPM8 antagonist AMTB (Figure 1e,f, Aubdool 2014). These results indicate that TRPM8 is involved, but at the time taking into account all the evidence that we had, we considered that this was a secondary role to TRPA1, as no further inhibition was observed in TRPA1KO mice when in the presence of AMTB. The concept that the TRPM8 component becomes important in ageing is novel to this manuscript and shown by use of the selective TRPM8 antagonist. We now try to make this reasoning clearer in the manuscript. (Please see discussion, particularly paragraph 4 onwards and line 663).

4) The proposed hypothesis that the lack of TRPM8 contribution in aged mice is somehow contradicted by the observation that the vasodilating response to menthol was reduced but not absent in aged mice. Given that the vasodilatation elicited by CA is presumably mediated by CGRP, which is the mediator of TRPM8-evoled vasodilatation? Could the reduced vasodilatation that follows TRPM8/TRPA1 activation somehow affect the vasoconstriction? In this line, as TRPA1 on sensory nerve terminal releases CGRP, it would be of interest to exclude that CGRP inhibition does not affect the vasoconstrictor response to cold.

Yes, this is a good point. We have previously shown that there are two distinct phases to the cold-induced vascular response and that whilst CGRP has a clear role in the relaxation/recovery phase, it does not play a role in the constrictor response. This we have shown via two experimental approaches in the Aubdool et al., 2014 manuscript. The first is that a CGRP antagonist has no involvement in the constrictor response (Figure 4d Aubdool et al., 2014), the second is that sensory nerve depletion with resiniferatoxin-treated mice (Figure 4f Aubdool et al., 2014), showed that if CGRP was blocked, or sensory nerves-depleted, a similar constrictor response as observed in vehicle- treated mice. (Please see Discussion section line 708-710).

5) It is explained that the cold-induced vasomotor response is present even after the chemical ablation of sensory fibers (ln. 651-653), therefore the age-dependent reduction in expression of TRPM8 and TRPA1 channels develops probably in sympathetic nerves. However, since expression of the channels in sensory nerves was not studied in the aging mice, it cannot be firmly excluded that an age-associated impairment in their cold-detecting function on afferent fibers also contributes to the attenuation of the vascular response. The authors should consider to discuss this possibility.

Thank you for this valuable comment. We have studied TRPM8 and TRPA1 mRNA expression in sensory DRGs via qPCR and for protein with WB too for TRPM8 (Figure 4g-i). In both cases TRPM8 levels were reduced in DRGs from aged mice. As stated in the manuscript there is not a TRPA1 antibody that passes standard scrutiny tests. We are continually testing antibodies, as we obtain them (including this month). Hopefully this is now written in a clearer manner in the manuscript. (Please see result section line 192-193).

6) In cases when the changes in a parameter are presented over time (Figures 1c, 2a, d, g, 3i, 5a ,d, g, and 6a,c), the authors should consider to run 2-way ANOVA with a post hoc test. Interestingly, the use of ANOVA is described in the text (ln. 791), but the results are not reported. In several of the aforementioned figure panels, the standard errors are missing from the curves, which should be corrected.

Figure 1c, 2a, d, g, 5a, d, g and 6a, c – These graphs show the raw data of blood flow, oxygen saturation and temperature trace graphs, and we have not performed any statistical analysis on them either with time or between young and aged. The purpose of these graphs were to show the mean data for blood flow, oxygen saturation and temperature graph for these experiments, thus we have not performed statistical analysis. However, we calculate statistical differences for the main parameters between young and aged mice that we were interested in for these graphs, based on our hypothesis. Thus we do so for maximum reduction/increase in blood flow, temperature change, and oxygen saturation change after cold treatment and the time it takes for blood flow to recover after cold treatment and we have used the raw data (Figure 1c, 2a,d,g, 5a,d,g, and 6a,c) to calculate that with the area under the curve (AUC), maximum reduction, and maximum increase value and then performed appropriate statistical analysis on them.

We have now added the standard error of mean on the graphs that were missing it. Please see figure 1c, 2a and 2g.

Figure 3i – is the summary figure of the cold plate test where the comparative response time is visualized by comparing with the difference in latency between all the temperatures tested in young and aged mice. We did this as Figures3a-c have different y-axis, due to the different response times at the different temperatures. The main purpose of graph 3i was illustrative only, to show that the difference in response time between young and aged mice is greatest at 20^o^C temperature. We did not originally carry out statistical test on this figure as our original hypothesis to be tested was not to compare response time of mice at different temperatures. The statistics for the original results (t-test) are shown in Figure 3a-c in keeping with our primary objective to investigate the difference is response time between young and mice at specific temperatures. We believe that statistically what we have done is correct, although logically we can understand why it would be nice for the viewer to see the stats on Figure 3i, based on the secondary hypothesis that the response time is important to analyse when compared with the overall temperatures examined. Therefore we have now included the statistical analysis and clarified in the text that this is a secondary hypothesis. We now clarify this logic in the text (lines 168-172).

7) The authors describe three different methods of anesthesia (ln. 691-696). It should be clarified which method was used in which experiments and, if necessary, explain whether different methods of the anesthesia could lead to differences in the results. A related question requiring explanation is that if medetomidine was used as part of the anesthesia (ln. 692), could it interfere with the effects of intraplantar medetomidine (or noradrenaline) used during the experiments (ln. 714)?

Thank you for pointing out this point and yes, it was necessary to change the anaesthesia to suit the protocols. The use of recovery anaesthesia allowed us to maximize the use of valuable aged mice. Please note, we used isoflurane as recovery anaesthesia because we were testing the effect of medetomidine on the blood flow response (Results 6a-d) (Lines 762-764 and 784-785). However, the use of isoflurane as anaesthesia requires continuous inhalation hence the use of a mask on the mouse was not possible with the cold water-hind paw blood flow experiment (as we had to move the mouse to dip its hind paw in the cold water) and agonist on ear-blood flow experiments (as we were measuring blood flow on the ear of the mice where the mask sits). Due to the nature of mouse handling required for these two experiments, we needed a recovery anaesthesia that did not restrict the movement of mouse during the protocol, thus we used ketamine/xylazine as our recovery anaesthesia. We have now made this clearer in methodology section (line 760-762).

8) Shivering thermogenesis belongs to the autonomic thermoeffectors instead of behavioral responses. The sentence in the Introduction should be corrected accordingly (ln. 54-58).

Thank you for pointing out this error. We have now corrected it. (Please see line 56-57).

9) Instead of the expressions "cool/cold temperatures", the authors should consider to use, for example, low/decreased/reduced temperature.

Yes, we thank the referees for this comment and now have made this change. However, as cool and cold provides clarity between temperatures that are specific to TPRA1 (cold) and TRPM8 (cool), we have tried retained the cold/cool terms after defining them in the context suggested. (Please see lines 82-87).

10) The legend of Figure 2d (ln. 345-346) should be revised: "mean blood flow recordings of hindpaw skin temperature" does not seem to make sense.

These were the mean blood flow readings. We have now put the s.e.m. on as suggested above and rewritten the legend revised legend to Figure 2 (Please see line 352).

11) The authors state that brown adipose tissue was collected from the mice for PCR (ln. 750-751), but data are not reported. Please clarify.

Apologies, this is a mistake, we have deleted these comments. (Please see line 824-825).

12) The use of abbreviations should be revised throughout the manuscript. Some abbreviations are not defined at first mention (e.g., NHS, WT, RT), while others are spelled out multiple times (e.g., CA). The reference to Σ-Aldrich should be unified, because, for example, in the Western blot section three different versions are used.

Thank you for these comments. We have done this, as required, throughout the manuscript.

We thank the referees and thank the Editorial Board for the chance to revise our manuscript.